# Compatibilization and Characterization of Polylactide and Biopolyethylene Binary Blends by Non-Reactive and Reactive Compatibilization Approaches

**DOI:** 10.3390/polym12061344

**Published:** 2020-06-14

**Authors:** Jose M. Ferri, Daniel Garcia-Garcia, Emilio Rayón, Maria D. Samper, Rafael Balart

**Affiliations:** Technological Institute of Materials (ITM), Universitat Politècnica de València (UPV), Plaza Ferrándiz y Carbonell 1, 03801 Alcoy, Spain; joferaz@epsa.upv.es (J.M.F.); dagarga4@epsa.upv.es (D.G.-G.); emraen@upvnet.upv.es (E.R.); rbalart@mcm.upv.es (R.B.)

**Keywords:** polylactide, biopolymers, blends, compatibilizing agents, microscopic techniques

## Abstract

In this study, different compatibilizing agents were used to analyze their influence on immiscible blends of polylactide (PLA) and biobased high-density polyethylene (bioPE) 80/20 (wt/wt). The compatibilizing agents used were polyethylene vinyl acetate (EVA) with a content of 33% of vinyl acetate, polyvinyl alcohol (PVA), and dicumyl peroxide (DPC). The influence of each compatibilizing agent on the mechanical, thermal, and microstructural properties of the PLA-bioPE blend was studied using different microscopic techniques (i.e., field emission electron microscopy (FESEM), transmission electron microscopy (TEM), and atomic force microscopy with PeakForce quantitative nanomechanical mapping (AFM-QNM)). Compatibilized PLA-bioPE blends showed an improvement in the ductile properties, with EVA being the compatibilizer that provided the highest elongation at break and the highest impact-absorbed energy (Charpy test). In addition, it was observed by means of the different microscopic techniques that the typical droplet-like structure is maintained, but the use of compatibilizers decreases the dimensions of the dispersed droplets, leading to improved interfacial adhesion, being more pronounced in the case of the EVA compatibilizer. Furthermore, the incorporation of the compatibilizers caused a very marked decrease in the crystallinity of the immiscible PLA-bioPE blend.

## 1. Introduction

Our society must face a difficult problem related the huge amounts of plastic products consumed worldwide. This situation is very marked in the packaging sector, with a wide range of single-use products that have contributed to a rapid increase in the amount of plastic waste in both controlled and uncontrolled landfills. Recycling of these wastes is very important for waste reduction and the subsequent positive effect on the carbon footprint of these materials. Nevertheless, the best way to minimize the use of petroleum-derived polymers is to replace them with other environmentally friendly alternatives. There is a trend toward the use of biodegradable (disintegrable in compost) and/or bio-based polymers; however, their cost is still high as they require important investments. Moreover, most of them are not easy to process. All of the above limit their use compared to the widely used petroleum-derived plastics.

Polylactide (PLA) is biodegradable, capable of disintegrating into a controlled compost with a standardized soil composition, and is a biocompatible polyester with high resistance and high transparency, but the use of PLA is limited due to its intrinsic high fragility and low toughness. Properly modified, PLA is used in different applications such as packaging, 3D printing, automotive and construction products, textiles, and biomedical applications, among others [1,2,3]. To overcome or minimize the disadvantages of PLA, different approaches can be carried out such as blending, plasticization, or copolymerization. Blending is the most feasible option since it provides a combination of clear cost-effectiveness balanced with the final properties. Blending consists of mixing two polymers in the melt state and cooled down to obtain mixtures with better or intermediate properties than both neat polymers. Blending of PLA with other polymers has been extensively analyzed using other biopolymers such as starch [4,5,6], polyhydroxyalkanoates (PHA) [7,8,9], poly(ε-caprolactone) (PCL) [10,11,12], or polybutylene adipate*-co-*terephthalate (PBAT) [13,14,15]. In addition, the effectiveness of blends of PLA with some petroleum-derived commodity plastics such as polyethylene terephthalate (PET) [16,17], polyvinyl chloride (PVC) [18], thermoplastic elastomers (TPEs) [19,20], polypropylene (PP) [21,22], and polyethylene (PE) [23,24,25,26] has been reported. Bearing in mind this wide potential of PLA blends, it should be noted that currently, PE can either be obtained from a petrochemical route or a bio-based approach, leading to so-called green polyethylene or bioPE. BioPE combines its natural origin with an easy processability, almost identical to its petroleum-derived counterpart. The main disadvantage of blending PE with PLA is the lack of miscibility, since PE is a non-polar polymer, and PLA is a polar polymer due to the presence of ester groups and oxygen-based groups in its structure [23]. This difference in polarity means that the solubility parameter (δ) of these polymers will be very different. In fact, the typical δ value for PLA is 20 MPa^1/2^ [27,28], while PE shows a δ value around 16.5 MPa^1/2^ [29], which indicates the low miscibility between them.

To improve the lack of miscibility in a blend, the use of compatibilizing agents is generalized. This system is the so-called non-reactive compatibilization. This technique is based on the use of compatibilizing agents, generally grafted or block copolymers, which are designed to chemically interact with the base polymers of the blend. This is possible due to the dual functionality that copolymers can provide, and thus the compatibilizing agent can establish interactions with both polymers in an immiscible blend, thus leading to the formation of bridges between both polymers with a positive effect on their partial miscibility. As a result, the mechanical properties of the blends are improved, and the interfacial adhesion is also improved. This is due to the fact that the typical immiscible blend shows a droplet-like morphology. The dispersed polymer phase is typically spherical and there is a gap between these droplets and the surrounding matrix. Compatibilizers positively contribute to reducing the domain sizes of the dispersed phase, and consequently, improving the morphological stability. Investigations of PLA/PE blends typically use low density PE (LDPE) and linear LDPE (LLDPE). To reduce the immiscibility, different compatibilizers have been proposed. Wang et al. [30] and Anderson et al. [31] synthesized PE-PLA di-block copolymers (PE*-b-*PLLA) as a compatibilizer. Other compatibilizers reported in the literature are maleic anhydride grafted PE (PE*-g-*MA) [23,25,32], ethylene-glycidyl methacrylate copolymer (EGMA) [33,34], and ethylene-methyl acrylate-glycidyl methacrylate (EMA-GMA) terpolymer [34,35]. The use of these compatibilizers plays a key role in reducing domain sizes and improving interfacial adhesion and, subsequently, mechanical properties. Other compatibilizing agents reported in the literature in PLA/PE blends have given interesting results. It is worthy to note the use of ethylene-glycidyl methacrylate-methyl acrylate terpolymer (PEGMMA) to improve miscibility with PP [36], and maleinized linseed oil (MLO) to improve PLA-TPS interaction by a combined plasticization–compatibilization effect [5]. This dual effect was also reported by Quiles-Carrillo et al. [37] by using MLO, epoxidized linseed oil (ELO), epoxy styrene-acrylic oligomer (ESAO), and styrene-glycidyl methacrylate copolymer (PS-GMA) as compatibilizing agents for blends of polyamide 1010/PLA, providing the best balanced mechanical properties.

Another interesting technique to improve miscibility between two immiscible polymers is the use of reactive extrusion (REX). This technique is carried out during compounding in the extruder. This technique requires the use of functionalized compatibilizers, characterized by reactive groups such as maleic anhydride, isocyanate, epoxide, and oxazoline, among others. At the same time, they are fed into the extruder, and a suitable initiator, usually a low-decomposition temperature peroxide (e.g., dicumyl peroxide, DCP) is also added. This rapidly decomposes and promotes polymer chain activation (free radical formation) along both polymers in the immiscible blend. These active points can react with the compatibilizers and hence, compatibilization takes place by reaction in the extruder. The obtained compound after a REX process consists of a complex structure with grafted polymers with compatibilizers at the interface. Therefore, the interfacial tension decreases, and the interfacial adhesion improves. This phenomenon allows obtaining a finer and more uniform droplet-like morphology. Consequently, the mechanical properties are usually improved since the gap between the two immiscible polymers is remarkably reduced. This somewhat allows interactions between both polymers and allows load transfer from the dispersed phase to the matrix phase, with a positive effect on the overall properties [38]. The use of DCP has been reported to improve the interaction of PHB/PCL [39], PLA/PBS [40], or thermoplastic dry starch (DTPS)/PLA using maleic anhydride (MA) as the compatibilizer and DCP as the peroxide initiator [41].

Bearing in mind the complexity of the PLA/PE system, and considering the increasing use of PLA in the packaging industry, PLA/PE blends could be a solution to recycle both polymers, thus the aim of this work was to analyze the influence of the newly proposed copolymers as compatibilizing agents (i.e., polyvinyl alcohol (PVA), ethylene-vinyl acetate (EVA) and reactive compatibilization (REX) with DCP) in PLA-bioPE blends in order to develop a material with balanced mechanical properties, and to study the influence of each compatibilizing approach on the morphology of the obtained blends by analyzing the size of the domains, the interface, and the mechanical properties.

## 2. Materials and Methods

### 2.1. Materials

The PLA-bioPE blends contained 20 wt % green PE and were obtained using Ingeo^TM^ Biopolymer 6201D commercial grade PLA, which was supplied by NatureWorks LLC (Minnetonka, MN, USA). Its density was 1.24 g cm^−3^ and contained about 1.5% D-isomer. Regarding the bioPE used in this study, it was a commercial grade SHA7260 high density biobased PE from Braskem supplied by FKuR (FKuR Kunststoff GmbH, Willich, Germany). This grade of HDPE is characterized by a minimum biobased polymer content of about 94% and a density of 0.955 g cm^−3^. Two non-reactive compatibilizers, namely PVA and EVA were used. PVA MOWIFLEX^TM^ C 17 and Repsol EVA Alcudia^®^ PA-461 with a vinyl acetate content of 33% was supplied by Ibiplast (Ibiplast, Ibi, Spain). Finally, a reactive compatibilization (REX) process using dicumyl peroxide (DCP) (98% purity) supplied by Sigma Aldrich (Madrid, Spain) was used. Figure 1 shows the chemical structure of the used compatibilizers.

### 2.2. Preparation and Processing

The sample preparation was carried out following the method described by Quiles-Carrillo et al. [23] with small modifications. First, PLA and PVA pellets were dried at 60 °C for 24 h before being processed to remove residual moisture due to the high sensitivity of polyesters to hydrolysis at high temperatures. Then, PLA (80 wt %), bioPE (20 wt %), and the different compatibilizing systems were manually mixed in a zipper bag to provide initial homogenization. The content of the compatibilizers and the code used can be seen in Table 1. After this initial homogenization, the different compositions were first processed in a twin-screw extruder with a speed of 60 rpm and a temperature profile between 172.5 and 180 °C. Subsequently, the material was processed in an injection molding machine Sprinter-11 (Erinca, Spain) at 180 °C to obtain 1BA standardized specimens for tensile tests and rectangular specimens (80 × 10 × 4 mm^3^) for flexural and Charpy impact tests. The injection molded samples were visually inspected, obtaining samples with a high homogeneity in terms of color and shape and the absence of surface defects.

### 2.3. Characterization Techniques

#### 2.3.1. Mechanical Properties

The mechanical characterization of the samples was carried out by tensile, flexural, and impact tests. Tensile tests were carried out at room temperature with an IBERTEST ELIB 30 (S.A.E. Ibertest, Madrid, Spain) machine following ISO 527. Tests were performed in 1BA standardized specimens with a crosshead speed of 5 mm min^−1^ using a load cell of 5 kN. At least five different samples of each composition were tested and average values of elongation at break, tensile strength, and elastic modulus were calculated.

Flexural tests were carried out using an IBERTEST ELIB 30 (S.A.E. Ibertest, Madrid, Spain) machine. All tests were performed at room temperature at a crosshead rate of 2 mm min^−1^ with a 5 kN load cell. At least five specimens from each composition were tested for the determination of flexural strength and flexural modulus. All tests were performed according to ISO 178.

To evaluate the ability of the different compositions to absorb energy, the Charpy impact test was carried out in a Charpy pendulum (1-J) from Metrotec S.A. (San Sebastián, Spain) following ISO 197:1993 on unnotched samples. At least five different specimens of each composition were tested, and average values were calculated.

#### 2.3.2. Dynamic-Mechanical Thermal Analysis (DMTA)

Thermomechanical characterization was performed by dynamic mechanical thermal analysis (DMTA) in shear-torsion mode using an oscillating rheometer model AR G2 from TA Instruments (New Castle, USA), equipped with a clamp system for solid samples. The rectangular samples had the following dimensions: 40 × 10 × 4 mm^3^. The test was carried out by scheduling a temperature ramp from 30 to 110 °C at a heating rate of 2 °C min^−1^; it was established to not exceed 110 °C in order to work with temperatures below the melting temperature of the studied materials and the melting temperature of these materials was determined by differential scanning calorimetry (DSC). All samples were tested at a frequency of 1 Hz and a maximum shear deformation (γ%) of 0.1%. These conditions were as described by Ferri et al. [42].

#### 2.3.3. Thermal Characterization

Differential scanning calorimetry (DSC) tests were carried out in a Mettler-Toledo 821 calorimeter (Schwerzenbach, Switzerland) in a nitrogen atmosphere (flow rate 66 mL min^−1^). Samples (around 5 mg) were introduced in standard aluminum pans. All tests consisted of a first heating stage from 25 °C to 200 °C to remove thermal history, followed by a cooling stage down to 25 °C, and finally a second heating up to 250 °C was used to evaluate all thermal transitions. The heating/cooling rate was 10 °C min^−1^ for all three stages. The glass transition temperature (*T*_g_), cold-crystallization peak temperature (*T*_cc_), and melting peak temperatures (*T*_m_) were determined using the information provided by the second heating scan, as previous thermal history had been removed. The degree of crystallinity (*χ_c_*) was calculated by using Equation (1), where Δ*H**_m_* is the melting enthalpy of *PLA*, Δ*H**_cc_* stands for the cold crystallization enthalpy of *PLA*, $\Delta H_{m}^{c}$ is the melting enthalpy associated with a theoretically fully crystalline *PLA*, reported to be 93 J g^−1^ [43], and *W**_PLA_*, stands for the actual weight content of *PLA* in blends.
(1)$$\chi_{c}\left(\%\right)=100\%\cdot \left[\frac{\Delta H_{m}-\Delta H_{cc}}{\Delta H_{m}^{c}}\right]\cdot \frac{1}{W_{PLA}}$$

#### 2.3.4. Field Emission Scanning Electron Microscopy (FESEM)

The morphology of PLA-bioPE blends was studied through surface characterization of the fractured samples from the impact tests. The morphologies were observed in a field emission scanning electron microscope (FESEM), ZEISS ULTRA 55 (Oxford Instruments, Pleasanton, CA, USA, EEUU) at an accelerating voltage of 2 kV. All samples were coated with a thin platinum layer using a high vacuum sputter coater model EM MED020 from Leica (Vienna, Austria).

#### 2.3.5. Transmission Electron Microscopy (TEM)

The study of the microstructure was carried out using a JEOL electron transmission microscope (TEM) model JEM 1010 worked at 80 kV. The samples were prepared using a Leica RM 2125RT (Vienna, Austria) ultra-microtome without pre-staining treatment.

#### 2.3.6. Nanoindentation

A G-200 nanoindenter (Agilent, Santa Clara, USA) was used to obtain the indentation elastic modulus (*E*) and hardness (*H*) of the prepared samples. Indentations were performed using a calibrated diamond Berkovich tip with a 30 nm radius. A 30-indentation array at a constant load of 650 mN (~2100 nm indentation depth) was done on the cross-section of the samples. When the analysis was done on single phases, indentations at a constant depth of 500 nm were programed. In these cases, tests were previously identified by an optical microscope installed into the nanoindenter. Samples were prepared by polishing with a grinder-polisher from Buehler Metaser (Beuhler UK LTD., Coventry, UK). Metallographic grinding papers of granulometry P240 and P1200 were initially used, and finally, samples were polished by using a ¼ µm diamond polishing compound.

#### 2.3.7. Atomic Force Microscopy (AFM) with PeakForce Quantitative Nanomechanical Mapping (QNM)

Height and stiffness maps were acquired on the cross-section of samples by a Veeco AFM under a PeakForce Tapping mode using Quantitative Nanomechanical Measurement, QNM [44,45]. A silicon cantilever with a force constant of 3.9 N/m was chosen to scan and indent the surfaces. The load was controlled to achieve a constant 5 nm contact depth using a previously calibrated tip of 2.5 nm radius. The scans were carried out onto 20 × 20 µm^2^ cross-section surfaces at 0.5 Hz. The elastic modulus was calculated by the Derjaguin–Muller–Toporov model, DMT [46], due to the observed adhesive contact between the sample and tip. Samples with shapes of thick slices were prepared by cutting the prepared formulations using an ultramicrotome, and subsequently kept on a dry silica cabinet. Before the experiments, samples were blown by nitrogen 5.0 to clean and dry the surfaces.

## 3. Results and Discussion

### 3.1. Mechanical Properties

Figure 2 gathers the main results obtained with regard to the mechanical characterization of the neat PLA samples, uncompatibilized PLA-bioPE blend, and compatibilized PLA-bioPE blends. Regarding the tensile mechanical properties, Figure 2a shows that PLA is a relatively brittle and rigid material characterized by a tensile strength of 69 MPa, an elastic modulus of 2.9 GPa, and a low elongation at break with a value of 6.3%. On the other hand, bioPE is a very ductile material, with a high elongation at break close to 520%, and a low tensile strength and elastic modulus [47]. This is partly due to the fact that at room temperature, it is far enough from its corresponding *T*_g_, which is located below −100 °C. After physical blending of PLA with 20 wt % bioPE, a significant decrease in tensile strength and elastic modulus is observed, with respective values of 49.4 MPa and 2.3 GPa for the uncompatibilized PLA-bioPE blend. This means a decrease of about 28% and 21% with respect to neat PLA. However, the elongation at break was hardly affected after the addition of bioPE, with values very similar to those of neat PLA. After the addition of the different compatibilizers in the PLA-bioPE blend, the tensile strength and the elastic modulus were hardly affected, obtaining values very similar to the uncompatibilized blend. In this case, the highest difference was found in the EVA-compatibilized blend, where a tensile strength of 46 MPa and an elastic modulus of 2 GPa was obtained, which means a respective decrease of 7% and 13% with regard to the tensile strength and elastic modulus of the uncompatibilized blend. Otherwise, relevant differences were observed in the elongation at break. As shown in Figure 2a, the addition of the different compatibilizers improved the elongation at break of the PLA-bioPE blend, resulting in an elongation at break of 9.7%, 13%, and 11.5% for the compatibilized blends with PVA, EVA, and DCP, respectively, which means an increase of 38.6%, 85.7%, and 64.3%, respectively with regard to the uncompatibilized blend (7%). This increase in the elongation at break on compatibilized blends is clear evidence of the effectiveness of the different compatibilizing agents used. All of them improve the compatibility, increasing the interaction between both polymers. In this case, the highest elongation at break was obtained for the sample compatibilized with 5 phr EVA. This is because the EVA copolymer has two co-monomers, ethylene (E), which leads to polyethylene segments or blocks, and vinyl acetate (VA), which polymerizes to give polyvinyl acetate segments or blocks. Polyethylene domains of EVA show a high degree of chemical affinity with the bioPE chains, and on the other hand, the polyvinyl acetate blocks of EVA, highly polar, provide good affinity with polar groups contained in PLA chains. This allows the EVA copolymer to act as a bridge between the two immiscible polymers, hence improving their compatibility and interfacial adhesion, reducing the stress concentration effect caused by the finely dispersed bioPE-rich phase [48,49].

A similar tendency can be observed with regard to the flexural properties. As can be seen in Figure 2b, the physical blend of PLA with bioPE led to a decrease in the flexural strength and elastic modulus due to the lack of compatibility between them, from 106.5 MPa and 2.9 GPa for neat PLA down to 72.8 MPa and 2.4 GPa for the uncompatibilized PLA-bioPE blend, representing a decrease of about 32% and 17%, respectively. After the addition of the different compatibilizers to the base PLA-bioPE blend, the flexural strength and modulus barely changed for the PVA- and DCP-compatibilized blends. However, the EVA-compatibilized blend showed a slight decrease in both flexural strength and flexural modulus, reaching values of 69.7 MPa and 2.1 GPa, respectively, which represents a decrease of 4.2% and 12.5%, respectively, regarding the uncompatibilized blend.

The impact-absorbed energy of each blend, which is representative for the toughness, is shown in Figure 2c. The energy absorption capacity is property related to material ductility, which is one of the most sensitive properties to the compatibility between the polymers in a blend since it depends notably on the material cohesion. In this case, neat PLA was characterized by a low energy absorption capacity, with a Charpy’s impact energy value of 16.1 kJ m^−2^, as it is a brittle material. The physical blend of PLA with bioPE resulted in a slight decrease in the energy absorption capacity with regard to neat PLA, leading to an energy absorption value in the PLA-bioPE blend of 13.8 kJ m^−2^, which represents a decrease of 14.3% compared to neat PLA. As with elongation at break, this is due to the lack of interaction between both polymers, which causes phase separation with low interfacial adhesion between them. This lack of interactions among the interface promotes a stress concentration phenomenon, thus leading to a more brittle material. Therefore, due to the lack of adhesion between the polymers, when the material is exposed to external stresses, microcracks are easily generated, and they can easily grow, thus reducing the energy absorption capacity of the material [50]. After the addition of the different compatibilizers, the impact-absorbed energy increased. In this case, the highest energy absorption was obtained for the EVA-compatibilized blend, reaching an energy absorption of 22.5 kJ m^−2^, which stands for an increase of nearly 63% with respect to the uncompatibilized blend. Similar results of impact strength in polyolefin PLA blends were also observed when compatibilizing agents by Zhou et al. [51] and Quiles et al. [23]. This increase in the energy absorption capacity of the blends with compatibilizers shows the effectiveness of all three compatibilizers used in this study, suggesting an improvement in the interaction between both polymers and, hence increasing their continuity. This allows load transfer along the interface, which has a positive effect on energy absorption.

After analyzing the mechanical properties, it can be concluded that the use of EVA as a compatibilizing agent in the PLA-bioPE blend had the best compatibilizing effect, which can be seen through a greater decrease in resistant mechanical properties such as tensile and flexural strength and a greater increase in ductile mechanical properties such as elongation at break and impact-absorbed energy.

### 3.2. Dynamical-Mechanical Thermal Analysis

The dynamic mechanical thermal analysis was complementary to corroborate the effectiveness of the different compatibilizers and their effect on the stiffness of PLA. In addition, DMTA was useful to understand their visco-elastic behavior in the analyzed temperature range.

The main changes in the storage modulus (G’) can be observed in Figure 3a. Between 60 to 80 °C, there was an abrupt change associated with the chain motions of PLA, which is related to the glass transition temperature (*T*_g_) of PLA. This can be obtained through the peak maximum of the dynamic damping factor (tan δ), as shown in Figure 3b. Specifically, for neat PLA, tan δ increases significantly during the glass transition process. The low percentage of crystallinity of neat PLA makes its amorphous phase responsible for the important increase in the viscous component in the *T*_g_ process, resulting in a very important softening of the material. Consequently, G’ of neat PLA is reduced by up to two orders of magnitude. The incorporation of bioPE, a remarkably more ductile polymer than PLA, to PLA results in a lower storage modulus compared to neat PLA. The simple fact of adding 20 wt % of bioPE reduced G’ from 1.65 GPa (neat PLA) to 1.39 GPa (at 30 °C), although no change in the *T*_g_ of PLA was observed. This is related to the lack of interfacial adhesion between PLA and bioPE [23]. Tan δ of PLA-bioPE blend is significantly lower than that obtained by neat PLA. This is mainly due to the viscous component of the complex modulus of PLA which decreases, while the elastic component increases because bioPE is in a rubbery state in that temperature range [52]. An increase in crystallinity in the PLA-rich phase due to the nucleating effect of bioPE could also contribute to achieving a lower tan δ.

On the other hand, the incorporation of compatibilizers had a very slight influence on the value of *T*_g_, obtained through the peak maximum of tan δ compared to the PLA-bioPE blend, showing differences depending on the type of compatibilizer. The use of DCP in the PLA-bioPE blend showed a slight improvement in the ductile properties due to a certain compatibilizing effect. Furthermore, the *T*_g_ value obtained from the tan δ peak temperature showed a decrease of only 1 °C, which verified its poor effect. However, the addition of PVA or EVA showed a greater compatibilizing effect, exhibiting slightly higher *T*_g_ decreases of 2 and 2.5 °C, respectively. Despite the decrease in *T*_g_ being very small for all three compatibilized systems, which indicates a slight compatibilizing effect, this was enough to provide increased ductile properties as above-mentioned; a decrease in the *T*_g_ of PLA in blends of polyolefin-PLA with compatibilized additives was also observed through DMTA characterization by Gao J. et al. [53] and Ma X. et al. [54]. This compatibility effect also had a clear effect on the storage modulus of the PLA-BioPE-EVA blend, going from a value of 1.39 GPa (PLA-BioPE blend) to 1.05 GPa, representing a reduction of 24.5% with respect to the PLA-bioPE formulation.

### 3.3. Thermal Characterization

DSC curve analysis is interesting to study the miscibility between two polymers. Figure 4 shows the DSC profiles (second heating ramp) of neat PLA, PLA-bioPE, and compatibilized PLA-bioPE. The main thermal parameters obtained from the DSC analysis of each sample are summarized in Table 2. The characteristic DSC curve for neat PLA shows three thermal transitions: the glass transition temperature (*T*_g_) around 62.7 °C, a cold crystallization process with a peak temperature (*T*_cc_) at 108.8 °C, and a melting process with a peak temperature (*T*_m_) around 171.8 °C. After the addition of 20 wt % bioPE, a second peak at about 131.1 °C could be detected in the PLA-bioPE blend, which is attributable to the melting process of bioPE (typical of a HDPE) as well as a slight decrease in the PLA melting peak temperature close to 2 °C. However, the *T*_g_ did not vary in a remarkable way, remaining practically constant. In general, the incorporation of bioPE into PLA maintains the melting temperature and *T*_g_ of pure PLA constant, which is a clear evidence of the poor interactions or the lack of miscibility between these two biopolymers [10]. The low miscibility between PLA and PE has also been observed through DSC characterization by Lu et al. [55], Zhao et al. [56], Quiles-Carrillo et al. [23], and Quitadamo et al. [57], which showed a similar behavior to that obtained in this work. Regarding the *T*_cc_, it can be observed that the presence of bioPE finely dispersed into the PLA matrix led to a noticeable decrease, close to 13 °C. This is due to the nucleating effect of bioPE on PLA, since at these temperatures bioPE is in a solid state, accelerating PLA crystallization [58].

As can be seen in Figure 4 and Table 2, the addition of the different compatibilizers to the PLA-bioPE blend hardly affected the *T*_g_ and *T*_m_ of PLA, obtaining values almost identical to those of the uncompatibilized PLA-bioPE blend. However, the *T*_cc_ of the PLA-rich phase increased slightly after the addition of the different compatibilizers to the PLA-bioPE blend. This increase in the *T*_cc_ in the compatibilized blends could be representative for the existence of a degree of compatibility between both polymers, which results in a restriction of the polymer chain mobility and therefore a slightly higher *T*_cc_ [58]. As shown in this case, the greatest increase in *T*_cc_ was observed in the EVA-compatibilized blend, where a *T*_cc_ of 102.2 °C was obtained, which was 6.6 °C higher than the uncompatibilized PLA-bioPE blend.

Regarding the degree of crystallinity, it can be observed in Table 2 that the crystallinity of the PLA-rich phase in the binary PLA-bioPE blend was substantially higher than that of the neat PLA. In particular, the degree of crystallinity (χ_c_%) of PLA was 13.4%, which is similar to the obtained values in previous research works [4]. After physical blending of PLA with 20 wt % of bioPE, the crystallinity of the PLA-rich phase increased to 52%. This increase in the crystallinity of the PLA-rich phase in the PLA-bioPE blend is due to the presence of bioPE domains, which increase the mobility of the PLA polymer chains, thus allowing the arrangement of the polymer chains during crystallization [59]. As can be seen, the addition of the different compatibilizers in the PLA-bioPE blend resulted in a remarkable decrease in the crystallinity of the PLA-rich phase with respect to the crystallinity of the PLA-rich phase of the uncompatibilized blend. In this case, the lowest crystallinity was reached in the EVA-compatibilized blend, in which a PLA-rich phase crystallinity of 25.3% was achieved, which means a decrease of about 51% with respect to the PLA-rich phase crystallinity in the PLA-bioPE blend. This decrease in crystallinity in the compatibilized blends may be associated with an increase in the compatibility between both polymers, which resulted in a lower mobility of the polymer chains affecting the crystal growth [60]. Furthermore, the reduction of the crystallinity of the PLA-rich phase of the compatibilized PLA-bioPE blends is in accordance with the decrease in the strength mechanical properties and the increase in the ductile mechanical properties with respect to the uncompatibilized blend.

### 3.4. Morphological Characterization

Figure 5 shows the FESEM images taken from the sample surface after the impact test. Figure 5a shows the fractured surface morphology of the neat PLA sample, showing a smooth surface corresponding to a typical brittle fracture surface. The fracture surface of the uncompatibilized PLA-bioPE blend (Figure 5(b1) and Figure 5(b2)) was completely different. In this case, it was possible to observe the immiscibility of both polymers by the formation of two well differentiated phases with a typical droplet-like structure. The matrix phase corresponds to the polymer with the highest proportion in the blend (i.e., PLA) and the dispersed phase is observed in the form of spherical domains, with a size range between 8–12 µm, generating an “island-and-sea” morphology, which corresponds to bioPE. The immiscibility between PLA and bioPE can also be seen by the poor interfacial adhesion that exists between the two phases. As shown in Figure 5(b2) (uncompatibilized PLA-bioPE sample), there was a small gap between the dispersed bioPE phase and the surrounding PLA matrix, resulting in a poor (or lack of) stress transfer between these two polymers in the blend. These gaps act as stress concentration points that cause a decrease in mechanical properties, making the material more fragile, as described above. The lack of miscibility between the two polymers is also evidenced by the presence of voids in the fracture surface corresponding to the pulled-out bioPE droplets by the impact test. The same behavior was observed by Vrsaljko et al. [61] for the PLA/LDPE blends.

The addition of the different compatibilizers into the blend (Figure 5c–e) resulted in a size reduction of the spherical bioPE domains, obtaining a size range between 2–8 µm. Furthermore, in the high magnification images of the compatibilized blends (Figure 5c–(e2)) it can be seen how the presence of compatibilizing agents in the PLA-bioPE blend leads to an increase in the interfacial adhesion between both polymers. This was observed by a more intimate bond between both phases and a reduction in the gap between them. The same behavior was observed by Quiles et al. [23] or Brito et al. [34]. This phenomenon would explain the improvement in the energy absorption capacity in the impact test, and in the increase in elongation at break on compatibilized blends.

### 3.5. Transmission Electronic Microscopy

Figure 6 shows the TEM images of the uncompatibilized and compatibilized PLA-bioPE blends. In these images, the applied force to obtain ultrathin slices by the microtome caused a deformation of the dispersed phase that corresponded to the flexible bioPE; as all slices were obtained in the same conditions, therefore the damage produced in the samples was similar in all. This deformation is a direct consequence of the lack of interaction between the PLA matrix and the dispersed bioPE phase. On the other hand, it can also be observed in the TEM images taken at 1000 magnification that the PLA-bioPE blend showed some dispersed HDPE domains, which were larger than those generated when using the different compatibilizers, PVA, EVA, and DCP. The same behavior was observed by Brito et al. [34], who observed a decrease in bioPE droplets when using ethylene-glycidyl methacrylate and ethylene-methyl acrylate-glycidyl methacrylate as compatibilizers for the bioPE-PLA blends. This conduct is identical to that observed by FESEM characterization. Furthermore, in the images taken at 2500× and 5000×, it can also be seen that the compatibilizing agents caused better adhesion of the dispersed phase with the matrix phase, since the gaps generated during microtomy preparation were smaller and the PLA-bioPE-EVA sample showed many domains without detachment. These findings confirm that EVA is the best compatibilizer of all three compatibilizing systems used in the PLA-bioPE blends. In accordance with previous mechanical characterization, an increase in ductile properties such as elongation at break and energy absorption by impact as well as improved interfacial adhesion, which was also demonstrated by FESEM.

### 3.6. Nanoindentation

To an in-depth analysis of the influence of compatibilizers on the mechanical properties of the individual phases of each formulation, the elastic modulus (*E*) and hardness (*H*) were obtained by the nanoindentation technique. The PLA material was individually analyzed within the PLA-bioPE formulation instead of the as-delivered. This was decided to consider the higher crystallinity of the PLA only for this formulation, as detected by DSC. The nanoindentation results are gathered in Figure 7. Results for the neat PLA matrix phase were *E* = 6 GPa and *H* = 500 MPa, higher than the expected values for this material, even considering its high degree of crystallinity. When a blend of PLA and bioPE polymers was analyzed by nanoindentation, the mechanical properties dropped down to *E* = 3.5 GPa and H = 200 MPa, hence corroborating the provided ductility by the less rigid bioPE phase. These results follow the rule of mixtures for a bi-phasic material.

However, the results acquired by the PLA-bioPE-compatibilized blends revealed lower averaged values, *E* = ~3 GPa and *H* = 200 MPa, corroborating the tendency described in previous section regarding tensile and flexural tests. Nevertheless, the sample containing EVA compatibilizer revealed a value of *E* = 2.5 GPa and *H* = ~100 MPa, slightly lower than that obtained by the other two compatibilizing systems. The improved cohesion between phases was contradictory with the lower value of *E* and *H* found in the PLA-bioPE-EVA formulation by nanoindentation. That is, while higher impact energy and elongation values were explained by a somewhat compatibilizing effect and better cohesion between phases, we think that the lower value of *E* and *H* acquired by nanoindentation in the PLA-bioPE blend with EVA, may be due to another phenomena such as the lower degree of crystallization of PLA, or perhaps other plasticizing effects due to the compatibilizer itself. That is, greater coherence between phases should result in higher *E* values by nanoindentation and not the displayed result. To clarify this ambiguous response, an AFM-QNM analysis was carried out to ascertain the actual effect of EVA.

### 3.7. Quantitative Nanomechanical Measurement Picoindentation (AFM-QNM)

To find additional evidence of the improved miscibility and relate them to changes in the mechanical properties due to the microstructure or crystalline changes by the effect of additives, the formulations were characterized by picoindentation using an AFM-QNM. In addition, an attempt was made to reveal the possible existence of a plasticizing effect that would explain the decrease in *E* values observed by nanoindentation in the case of the PLA-bioPE-EVA formulation. Figure 8 shows the height channel (upper sequence) and the elastic modulus channel (lower sequence) of the studied blends. The cohesion failure between phases was corroborated for the PLA-bioPE system. This bonding failure was also found in the sample with PVA, although to a lesser degree.

The images of the PLA-bioPE sample also revealed a microstructure of big droplets of bioPE compared with the smaller bioPE droplets found in the PLA-bioPE compatibilized blends. The droplet size decrease was due to the improved miscibility between the PLA and bioPE phases. The finest droplet size decrease was found in the EVA-compatibilized blend. This phenomenon could also be seen in the FESEM images, but was clearly revealed by AFM due to its preparation by ultra-microtomy. Furthermore, in the height channel, the PLA-bioPE showed a smoother surface than the compatibilized blends, which revealed bulges and waves. This roughness produced by the dragging of the microtome only in the compatibilized blends indicates an increase of the plasticization feature, and can be explained by the lower crystallinity and to other physicochemical changes that could be occurring.

Nevertheless, in the stiffness channel of Figure 8, a mechanism of dragging where the bioPE is sliced toward the PLA was revealed. This sliced material was caused by the friction with the blade of the ultra-microtome and is related to an increase in the plasticization capacity in both phases, being more notably visualized by the bioPE. This plasticization could be due to a decrease in PLA crystallinity, as commented before, or perhaps due to changes in the physicochemical properties of the bioPE. Secondary undetected products generated by the compatibilizer are also possible. The colors were more contrasted for the PLA-bioPE sample due to the higher crystallinity of PLA, provoking a higher modulus values between phases.

Putting all these results together, it can be concluded that there are several factors involved in the resulting mechanical response of the compatibilized PLA-bioPE blends. Compatibilizers improve the cohesion between phases, raising the elongation at break and impact-absorbed energy. However, the droplet boundary, which blocks the plastic deformation mechanisms also increased by the compatibilizers, revealed a better droplet-like morphology with finer droplet sizes. Furthermore, the plasticization phenomena explained by the reduction of the crystallinity and other overlapping phenomena could suggest that the mechanical response of the material must be a complex balance between all the commented factors.

## 4. Conclusions

The present study evaluated the use of different compatibilizers to improve the interaction of PLA and bioPE in binary blends, since both polymers are incompatible due to their different polarities. Specifically, blends containing 80 wt % PLA and 20 wt % bioPE were analyzed. The resulting blend offers a significant loss of mechanical properties regarding neat PLA. The morphology revealed by different microscopic techniques showed the typical droplet-like or “island-in-the-sea” structure in which the minor component (bioPE) was finely dispersed in the form of spherical droplets with small gaps with the surrounding PLA matrix, leading to a lack of interaction.

The compatibilizing agents used in this work were PVA and EVA (non-reactive compatibilization) and DCP (for reactive compatibilization). The use of these compatibilizers in the PLA-bioPE blend improved the interaction of both polymers since they increased the ductile mechanical properties such as elongation at break and impact-absorbed energy due to an increase in the cohesion of the polymers. This situation was much more pronounced when EVA was used as a compatibilizer, which increased the elongation at break and the impact absorption energy by 85.7% and 63%, respectively, with respect to the uncompatibilized blend. Furthermore, the improvement in the interaction of PLA and bioPE was ascertained by FESEM and TEM, observing a decrease in the bioPE droplet size and an improvement in the interfacial adhesion (reduction of the gap between the bioPE droplets and the surrounding PLA matrix), which corroborated the data obtained in the mechanical characterization.

On the other hand, the compatibilizing effect of the different compatibilizing agents used was also shown by DSC. In this case, it was observed how the addition of the compatibilizers in the PLA-bioPE blend led to an increase in the cold crystallization peak temperature. In addition, the degree of crystallinity was also affected. When both polymers, PLA and bioPE, were blended, there was a significant increase in the degree of crystallinity of PLA due to the nucleating effect produced by bioPE; however, when compatibilizers were added, a marked decrease in the crystallinity of the PLA-bioPE blend was observed, around 50%.

The mechanical response of the formulations studied by means of nanoindentation and AFM-QNM indicated that the properties of the blends will depend to a great extent on the improvement of cohesion and miscibility induced using compatibilizers. However, the following points should also be considered in the final balance of the mechanical behavior:(i)Increase in interfacial adhesion, which was the highest when EVA was used.(ii)The microstructure shows a droplet size reduction (bioPE), being more pronounced in the case of EVA.(iii)The 50% decrease in crystallinity due to the use of compatibilizers may affect the resulting mechanical properties.(iv)There was somewhat plasticization of both PLA and bioPE identified, which may be due to possible physical-chemical changes in the material caused by the effect of compatibilizers.

It can be concluded that the use of compatibilizing agents in PLA-bioPE blends is an attractive strategy to improve miscibility between these two biopolymers. The results showed an improvement in the ductile properties and a decrease in the droplet size of the disperse bioPE domains, which is representative for a better miscibility of both polymers. It should be noted that EVA is the most attractive compatibilizing agent of those studied in this work.

## Figures and Tables

**Figure 1 polymers-12-01344-f001:**
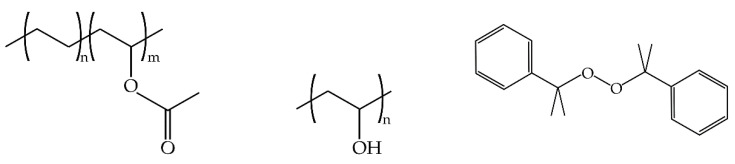
Representation of the chemical structure of different compatibilizers used in polylactide/biobased high-density polyethylene (PLA/bioPE) immiscible blends.

**Figure 2 polymers-12-01344-f002:**
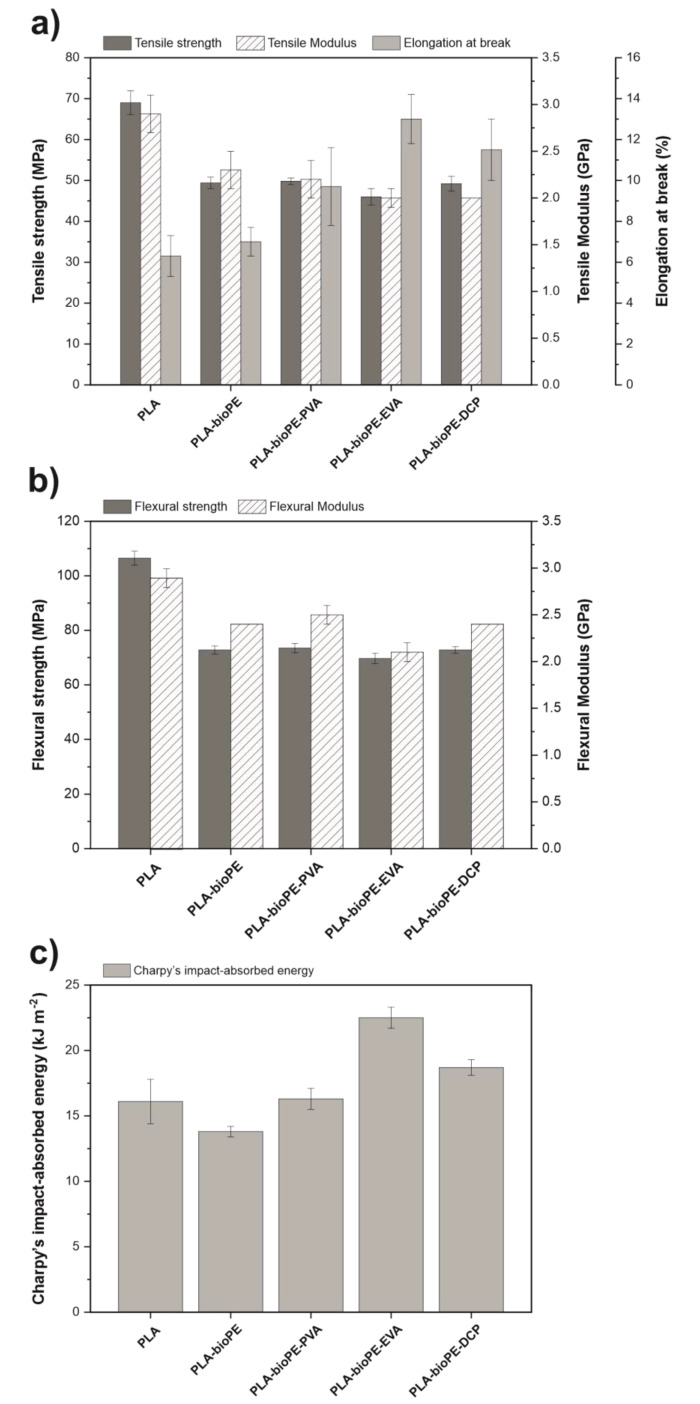
Mechanical properties of neat PLA, PLA-bioPE blend, and PLA-bioPE blend compatibilized with polyvinyl alcohol (PVA), polyethylene vinyl acetate (EVA) and dicumyl peroxide (DCP): (**a**) tensile properties; (**b**) flexural properties; and (**c**) impact properties.

**Figure 3 polymers-12-01344-f003:**
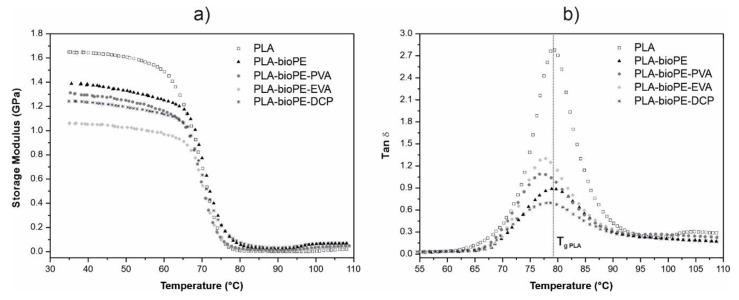
Comparison of the evolution of the (**a**) storage modulus (G’) and (**b**) dynamic damping factor (tan δ) for neat PLA, PLA-bioPE, and PLA-bioPE with different compatibilizers.

**Figure 4 polymers-12-01344-f004:**
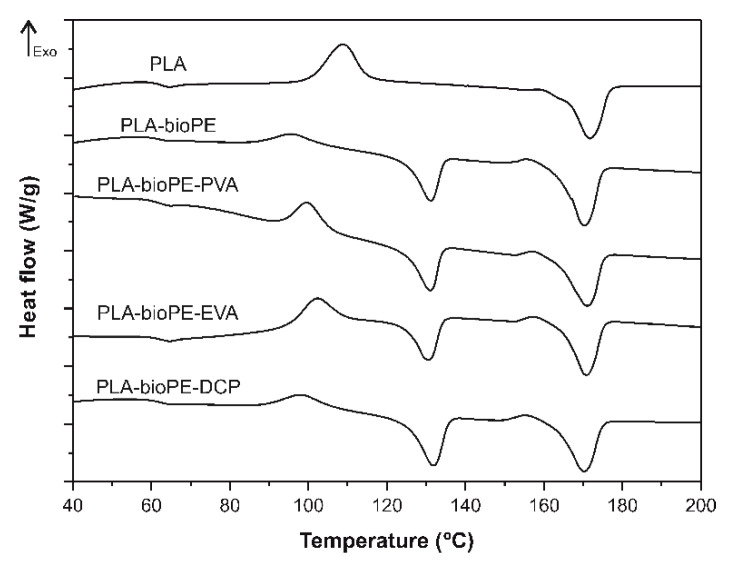
DSC curves of the second heating cycle of neat PLA, PLA-bioPE blend, and PLA-bioPE blend compatibilized with PVA, EVA, and DCP.

**Figure 5 polymers-12-01344-f005:**
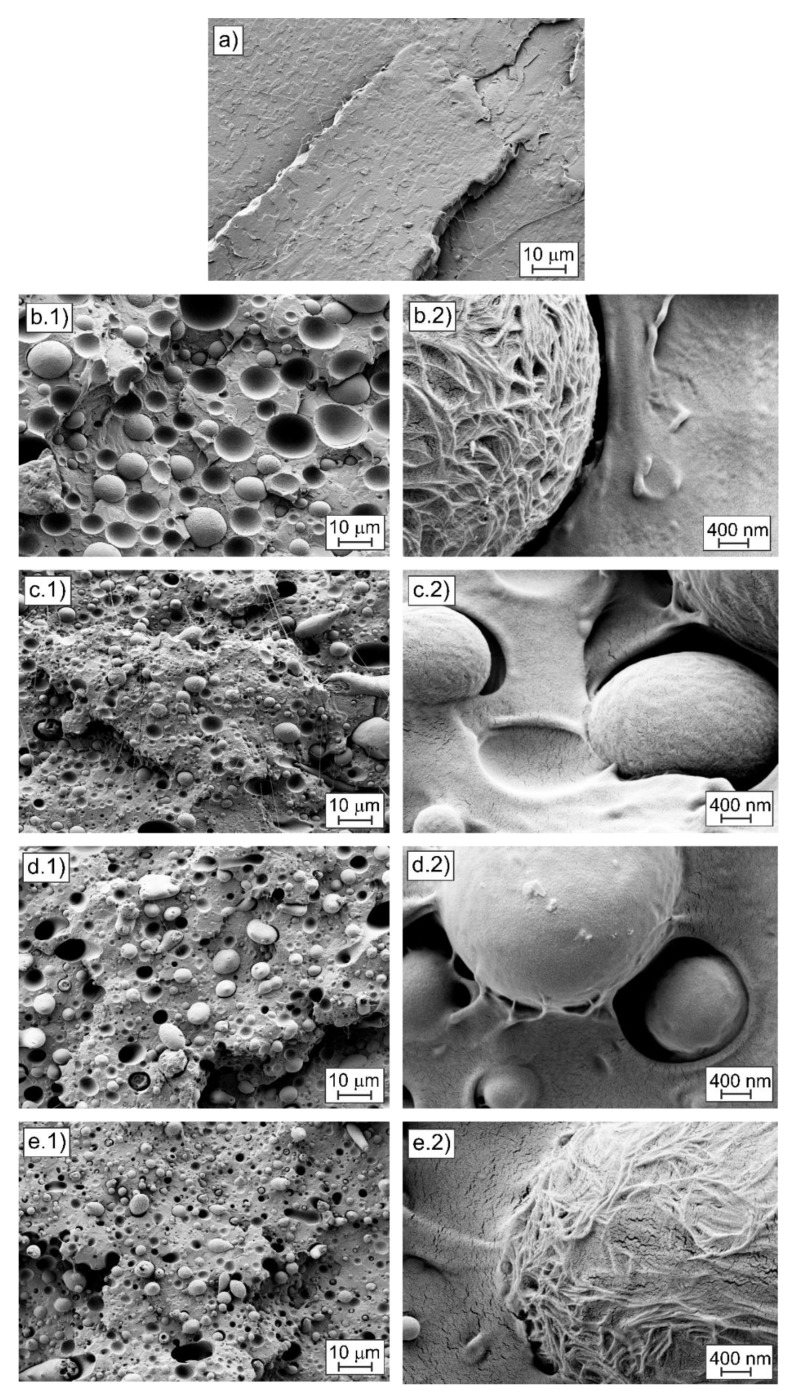
Field Emission Scanning Electron Microscopy (FESEM) images of the fracture surface from impact test of: (**a**) neat PLA at 1000×; (**b1**) PLA-bioPE at 1000× and (**b2**) PLA-bioPE at 25,000×; (**c1**) PLA-bioPE-PVA at 1000× and (**c2**) PLA-bioPE-PVA at 25,000×; (**d1**) PLA-bioPE-EVA at 1000× and (**d2**) PLA-bioPE-EVA at 25,000×; (**e1**) PLA-bioPE-DCP at 1000× and (**e2**) PLA-bioPE-DCP at 25,000×.

**Figure 6 polymers-12-01344-f006:**
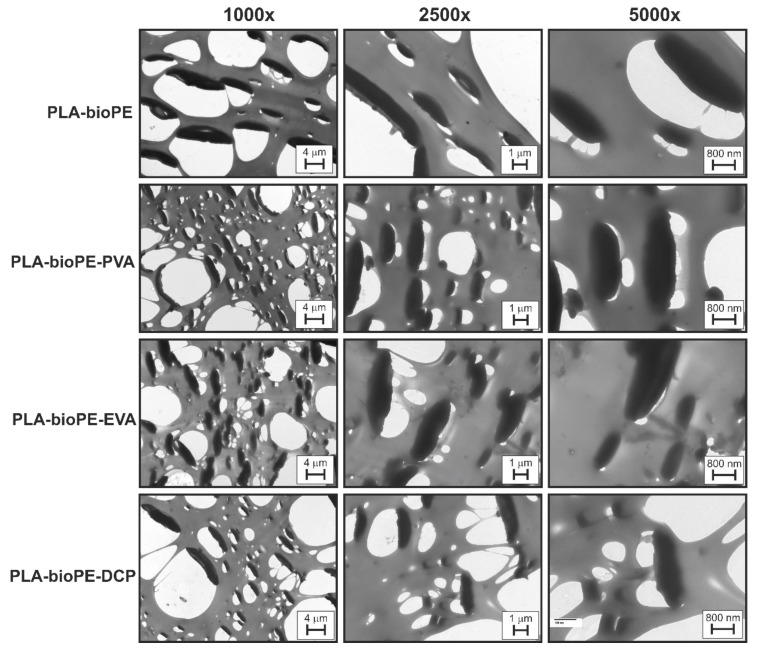
Transmission Electron Microscopy (TEM) micrographs at 1000×, 2500×, and 5000× of the PLA-bioPE blend and PLA-bioPE blend compatibilized with PVA, EVA, and DCP.

**Figure 7 polymers-12-01344-f007:**
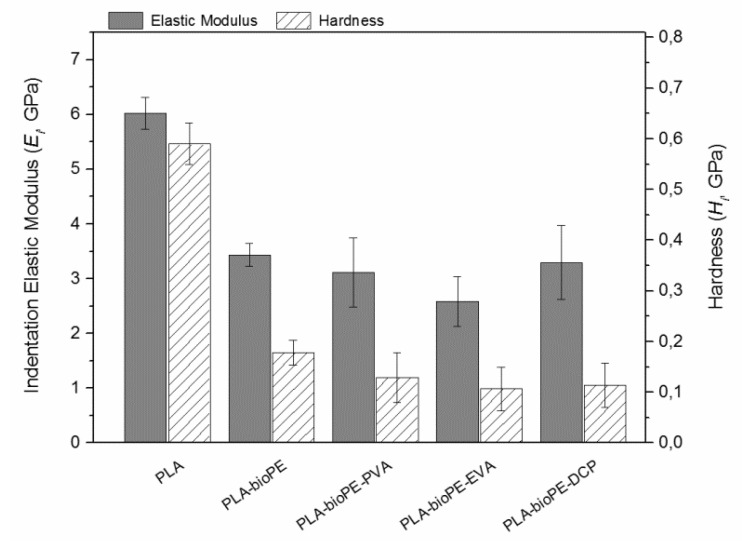
Nanoindentation E and H results of neat PLA, PLA-bioPE, and PLA-bioPE blend compatibilized with PVA, EVA, and DCP.

**Figure 8 polymers-12-01344-f008:**
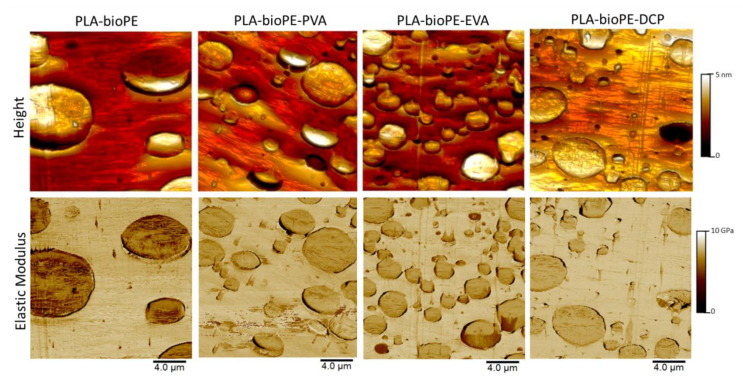
Height (superior) and elastic modulus (inferior) Atomic Force Microscopy maps obtained by Atomic Force Microscopy with PeakForce Quantitative Nanomechanical Mapping (AFM-QNM) on PLA-bioPE and PLA-bioPE blend compatibilized with PVA, EVA, and DCP.

**Table 1 polymers-12-01344-t001:** Code and composition of uncompatibilized and compatibilized polylactide/biobased high-density polyethylene (PLA/bioPE) blends.

Code	PLA (wt %)	bioPE (wt %)	PVA (phr *)	EVA (phr)	DCP (phr)
PLA	100	-	-	-	-
PLA-bioPE	80	20	-	-	-
PLA-bioPE-PVA	80	20	5	-	-
PLA-bioPE-EVA	80	20	-	5	-
PLA-bioPE-DCP	80	20	-	-	0.2

* phr (per hundred resin) represents the weight parts of the compatibilizers added to one hundred weight parts of the base PLA-bioPE blend.

**Table 2 polymers-12-01344-t002:** Differential scanning calorimetry (DSC) parameters of neat PLA, PLA-bioPE blend, and PLA-bioPE blend compatibilized with PVA, EVA, and DCP.

Samples	DSC Parameters							
	*T*_m PE_ (°C)	∆H_m PE_ (J g^−1^)	*T*_g PLA_ (°C)	*T*_cc PLA_ (°C)	∆H_cc PLA_ (J g^−1^)	*T*_m PLA_ (°C)	∆H_m PLA_ (J g^−1^)	X_c PLA_ (%)
PLA	-	-	62.7	108.8	26.6	171.8	−39.1	13.4
PLA-bioPE	131.1	−104.2	62.5	95.6	9.1	170.2	−47.8	52.0
PLA-bioPE-PVA	130.9	−105.5	62.4	100.1	19.2	170.8	−40.9	29.2
PLA-bioPE-EVA	130.7	−103.2	62.9	102.2	20.7	170.9	−39.5	25.3
PLA-bioPE-DCP	131.5	−125.5	62.4	98.1	11.2	170.3	−40.9	39.9

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
