# Peer review of "Compatibilization and Characterization of Polylactide and Biopolyethylene Binary Blends by Non-Reactive and Reactive Compatibilization Approaches"

_polymers, 2020, doi:10.3390/polym12061344_

Round 1
Reviewer 1 Report
There are several comments or concerns for the authors to take care of before the manuscript is published.
- Line 41: re-rewrite the sentence without using brackets, which makes the sentence harder to understand.
- Line 42: delete “properties”.
- Line 44: the comma seems wrong.
- Line 172-176: This experiment is wrong. The second cycle of DSC should not exceed the limit of the first cycle, i.e., 200oC under this circumstance. Also, TGA should be performed ahead of time to make sure that the sample would not decompose in the proposed temperature range. It is understood that PLA is so well studied that maybe the results would not make a difference but not always.
- Line 224: The correct expression is Tg rather than Tg. “g” should be subscripted. This is very important. Line 315 and line 320 are examples.
- The authors used a lot of redundant and unclear expressions such as the sentence from line 223 to line 225, i.e., “On the other hand, xxx (since xxx), which is xxx, close to 520%, as well as xxx.” Sentences like this appeared a lot in the manuscript and should be re-written.
- Somehow, Figure 3 is blurry. Micrographs with higher qualities are to be replaced with.
- Figure 3 and Figure 4: are there any way to make the tests in these two figures consistent, e.g., run modulus-temperature test also up to 200oC instead of 110oC and plot it together with the corresponded DSC curve?
Author Response
Reviewer 1
There are several comments or concerns for the authors to take care of before the manuscript is published.
- Line 41: re-rewrite the sentence without using brackets, which makes the sentence harder to understand.
ANSWER
As suggested by reviewer 1, the sentence has been modified in order to provide greater clarity.
“Polylactide (PLA) is a biodegradable, capable of disintegrating into a controlled compost with a standardized soil composition, and biocompatible polyester with high resistance and high transparency, but the use of PLA is limited due to its intrinsic high fragility and low toughness”.
- Line 42: delete “properties”.
ANSWER
As suggested by reviewer 1, the word “properties” has been deleted from the sentence.
- Line 44: the comma seems wrong.
ANSWER
Thanks for this comment. As the reviewer says, the comma was incorrectly placed, making it difficult to understand the sentence, so it has been removed.
- Line 172-176: This experiment is wrong. The second cycle of DSC should not exceed the limit of the first cycle, i.e., 200oC under this circumstance. Also, TGA should be performed ahead of time to make sure that the sample would not decompose in the proposed temperature range. It is understood that PLA is so well studied that maybe the results would not make a difference but not always.
ANSWER
Thanks for this comment. In this case two heating stages were carried out, a first cycle with the aim of removing the previous thermal history of the processed material, and a second cycle to obtain all the thermal transitions of the blends after a controlled cooling process at a constant cooling rate. In this case, a maximum temperature of 200 ºC was established for the first cycle, due to the fact that this temperature is slightly higher than the melting temperature of both polymers in the blend. The degradation temperature of both polymers was also taken into account to establish this temperature. In previous studies carried out by our research group, it was determined by TGA that PLA started to degrade at around 331 ºC [Reactive toughening of injection-molded polylactide pieces using maleinized hemp seed oil] while the initial degradation temperature of BioPE was around 470 ºC [Development of a biocomposite based on green polyethylene biopolymer and eggshell]. As can be seen, these degradation temperatures are significantly higher than the final temperature established in the first heating cycle, which is aimed at eliminating the thermal history of the processed samples. After removing the thermal history of the samples, a second cycle was performed in order to obtain all the thermal transitions of the blends. In this case it was decided to increase the final temperature to ensure that all thermal transitions took place correctly. The whole range of temperatures above the melting temperature of PLA is not taken into consideration for the present study, therefore it does not influence that the final temperature of the second cycle exceeds the temperature of the first cycle. In any case, the aim of each heating stage has been stated in the manuscript to avoid any confusion.
- Line 224: The correct expression is Tg rather than Tg. “g” should be subscripted. This is very important. Line 315 and line 320 are examples.
ANSWER
We totally agree with this reviewer’s comment, the manuscript has been completely revised and the incorrect expression Tg has been changed to the correct expression Tg.
- The authors used a lot of redundant and unclear expressions such as the sentence from line 223 to line 225, i.e., “On the other hand, xxx (since xxx), which is xxx, close to 520%, as well as xxx.” Sentences like this appeared a lot in the manuscript and should be re-written.
ANSWER
As reviewer 2 suggests, the above-mentioned sentence has been modified. In addition, as recommended by the reviewer, and in-depth check of the manuscript has been carried out, paying special attention to these redundant and unclear sentences and, accordingly, these have been modified to avoid confusion. Other sentences in the manuscript that were redundant, and confusing have also been modified.
- Somehow, Figure 3 is blurry. Micrographs with higher qualities are to be replaced with.
ANSWER
As reviewer 1 suggests, Figure 3 has been re-plotted aging in order to increase its quality. In addition, we will proceed to upload the image file in TIFF and JPEG formats to the platform, in case it is needed for final production.
- Figure 3 and Figure 4: are there any way to make the tests in these two figures consistent, e.g., run modulus-temperature test also up to 200oC instead of 110oC and plot it together with the corresponded DSC curve?
ANSWER
We thank the reviewer for this comment. In this work the DMTA test maximum temperature was selected on the basis of the results obtained by DSC of PLA-bioPE blend. For this reason, it was established not to exceed 110 ºC, since in DMTA with solid samples, it is always necessary to work at temperatures lower than the melting temperature to determine its thermo-mechanical properties in dynamic stress conditions. In this case, the maximum temperature has been set to 110 ºC, so that it is possible to observe the glass transition temperature and the cold crystallization temperature of PLA. It is important to bear in mid that the melt peak temperature of biPE is close to 130 ºC. Therefore, the selected maximum temperature for DMTA tests must be below 130 ºC as bioPE melts.
Reviewer 2 Report
The article is devoted to the application of the compatibilizing agents for co-polymerization of PLA with other non-polar polymers. In my opinion, the work has been correctly justified and can be found as valuable for readers involved in polymer science. The authors presented a lot of experimental techniques, good quality of presentation.
Undoubtfully, the article is worthy of publication, but after thorough language corrections. I’m not an expert in English, but even I see a lot of mistakes. A native speaker should extensively correct the text to make it easy-readable. Many words are misused.
I have no comments concerning the merit of the article
Author Response
Reviewer 2
The article is devoted to the application of the compatibilizing agents for co-polymerization of PLA with other non-polar polymers. In my opinion, the work has been correctly justified and can be found as valuable for readers involved in polymer science. The authors presented a lot of experimental techniques, good quality of presentation.
Undoubtfully, the article is worthy of publication, but after thorough language corrections. I’m not an expert in English, but even I see a lot of mistakes. A native speaker should extensively correct the text to make it easy-readable. Many words are misused.
I have no comments concerning the merit of the article
ANSWER
As Reviewer 2 suggests an in-depth revision of the English grammar and spelling has been carried out and all detected mistakes have been corrected in order to increase clarity and understanding. Some sentences have been rewritten to avoid confusion and all detected typos have been corrected.
Reviewer 3 Report
It is a very interesting work, however some questions should be done:
- Section 2.2 - Why the choice of the method used? Based on what? Previous works? Open literature? Standard?
- Section 2.3.2 – Based on what? Based on what were these test conditions chosen?
- Figure 2 (c) – Impact Energy? I do not think so.
- Figures of damage obtained after the tests? The type of damage can help to justify the results.
- The results obtained are expectable or not? Why?
- Were the specimens inspected before the tests? Are there any small defects in the specimens before they are tested?
- Without discussion with open literature, the conclusions can be questionable.
Author Response
Reviewer 3
It is a very interesting work, however some questions should be done:
- Section 2.2 - Why the choice of the method used? Based on what? Previous works? Open literature? Standard?
ANSWER
Thanks for the comment. In this case the method used to obtain uncompatibilized and compatibilizer PLA-bioPE blends has been widely used at both research and industrial scale. In fact, it is a standard way to obtain binary or ternary blends in the melt state. This standard includes drying, pre-mixing, compounding (extrusion) and injection moulding (samples for further characterization). A previous work developed by the research group has been provided to give support to this standard method of obtaining binary blends, with and without compatibilizers.
- Section 2.3.2 – Based on what? Based on what were these test conditions chosen?
ANSWER
Regarding to the DMTA test, temperature was selected on the basis of the results obtained by DSC of PLA-bioPA blends. For this reason, it was established not to exceed 110 ºC, since in the DMA test it is always necessary to work with temperatures lower than the melting temperature to determine its thermo-mechanical properties. The frequency and shear deformation conditions were used the same as in previous works, which have been referenced in the manuscript. It is worthy to note that bioPE melts at 120 – 140 ºC. Therefore, it is necessary that the maximum temperature in a DMTA does not reach 120 ºC. In this case, it has been selected a maximum temperature of 110 ºC for DMTA characterization as it allows obtaining the glass transition temperature and the cold crystallization process of PLA.
- Figure 2 (c) – Impact Energy? I do not think so.
ANSWER
We agree with the reviewer. “impact energy” is not the best name for this property. Accordingly, it has been changed to “Charpy’s impact-absorbed energy”, which represents in a clearer way the results shown in Figure 2c.
- Figures of damage obtained after the tests? The type of damage can help to justify the results.
ANSWER
This comment is really interesting. Prior to the visualization of the samples by TEM, we did not know that using microtomy would produce a deformation of the dispersed flexible phase (bioPE). Nevertheless, all samples for TEM characterization were obtained in the same way, applying the same force and, therefore, we consider that the damage produced in the samples is similar in all of them. The difference between samples is due to the improvement of the interaction of both phases, PLA and bioPE, thanks to the incorporation of the different compatibilizing agents.
- The results obtained are expectable or not? Why?
ANSWER
Due to the different polarity of both polymers, PLA and bioPE, we were aware of the incompatibility between them. For this reason, we decided to study the improvement of the interaction phenomena by using different compatibilizers. By adding the compatibilizers we expected to improve the properties of the 80%PLA-20%bioPE blend but we did not know the extent of compatibilization and which compatibilizer provided the best results. Therefore, this work assesses the usefulness of some compatibilizers with dual functionality to improve the interface interactions between the bioPE droplets and the surrounding PLA matrix.
- Were the specimens inspected before the tests? Are there any small defects in the specimens before they are tested?
ANSWER
As commented by reviewer 3, the samples were visually inspected after the injection moulding process, and before they were tested, and no surface defects were observed in the obtained samples. In addition, injection moulding allows to obtain samples with a high homogeneity in terms of shape and colour due to the high injection pressure. All of this is reflected in the manuscript.
- Without discussion with open literature, the conclusions can be questionable.
ANSWER
We agree with the reviewer about this comment. Therefore, we have provided additional secondary literature and compared the results reported in the open literature with those obtained in the present work. The following references have been added in different parts of the “Results & discussion” section to give support to the obtained results, and the reported results have been compared with the herein obtained results. These comments contribute to give support to the “Conclusions” section.
Round 2
Reviewer 1 Report
Questions were well addressed by the authors.
Reviewer 3 Report
After reading carefully the paper it is possible to see significant improvements.